# Oxidative Stress and Left Ventricular Performance in Patients with Different Glycometabolic Phenotypes

**DOI:** 10.3390/nu14061299

**Published:** 2022-03-18

**Authors:** Velia Cassano, Sofia Miceli, Giuseppe Armentaro, Gaia Chiara Mannino, Vanessa Teresa Fiorentino, Maria Perticone, Elena Succurro, Marta Letizia Hribal, Francesco Andreozzi, Francesco Perticone, Giorgio Sesti, Angela Sciacqua

**Affiliations:** 1Department of Medical and Surgical Sciences, University Magna Græcia of Catanzaro, 88100 Catanzaro, Italy; velia.cassano@libero.it (V.C.); sofy.miceli@libero.it (S.M.); peppearmentaro@libero.it (G.A.); gaiamannino@unicz.it (G.C.M.); vanessa.fiorentino@unicz.it (V.T.F.); mariaperticone@unicz.it (M.P.); succurro@unicz.it (E.S.); hribal@unicz.it (M.L.H.); andreozzif@unicz.it (F.A.); perticone@unicz.it (F.P.); 2Research Center for the Prevention and Treatment of Metabolic Diseases, University of Catanzaro, 88100 Catanzaro, Italy; 3Department of Clinical and Molecular Medicine, University Rome-Sapienza, 00185 Roma, Italy; giorgio.sesti@uniroma1.it

**Keywords:** oxidative stress, global longitudinal strain, type 2 diabetes, hyperglycaemia, organ damage

## Abstract

The aim of the present study was to evaluate the possible correlation between oxidative stress and subclinical myocardial damage, assessed with speckle tracking echocardiography (STE), in normal glucose tolerance (NGT) patients with one-hour plasma glucose values ≥ 155 mg/dL (NGT ≥ 155), comparing them to NGT < 155 subjects, impaired glucose tolerance (IGT) and type 2 diabetes mellitus (T2DM) newly diagnosed patients. We enrolled 100 Caucasian patients. All subjects underwent OGTT. The serum values of oxidative stress markers (8-isoprostane and Nox-2) were assessed with an ELISA test. Echocardiographic recordings were performed using an E-95 Pro ultrasound system. We observed significant differences, among the four groups, for fasting plasma glucose (*p* < 0.0001), one-hour postload (*p* < 0.0001), and two-hour postload plasma glucose (*p* < 0.0001). As compared with NGT < 155, NGT ≥ 155 exhibited significantly worse insulin sensitivity and higher values of hs-CRP. No significant differences were observed between NGT ≥ 155 and IGT patients. There was a significant increase in 8-isoprostane (*p* < 0.0001) and Nox-2 (*p* < 0.0001), from the first to fourth group, indicating an increase in oxidative stress with the worsening of the metabolic status. Serum levels of 8-isoprostane and Nox-2 were significantly increased in NGT ≥ 155 compared to the NGT < 155 group, but similar to IGT. The global longitudinal strain (GLS) appeared progressively lower proceeding from the NGT < 155 to T2DM group (*p* < 0.0001). For similar values of left ventricular ejection fraction (LVEF), NGT ≥ 155 exhibited reduced GLS compared to NGT < 155 (*p* = 0.001), but similar to IGT patients. Our study demonstrated that NGT ≥ 155 subjects exhibit early functional impairment of myocardial contractile fibres, these alterations are correlated with increased oxidative stress.

## 1. Introduction

In recent decades, many studies suggested that one-hour postload plasma glucose values ≥155 mg/dL (8.6 mmol/L), during an oral glucose tolerance test (OGTT), detects individuals at an increased risk of developing type 2 diabetes mellitus (T2DM) among those who have normal glucose tolerance (NGT) [1]. Individuals with one-hour postload plasma glucose ≥ 155 mg/dL (NGT ≥ 155) show a worse metabolic and hemodynamic profile, which contributes to an increased cardiovascular (CV) risk [2], as compared with individuals with one-hour postload plasma glucose < 155 mg/dL (NGT < 155). One-hour postload plasma glucose levels during OGTT are strongly related with subclinical CV organ damage including increased carotid intima-media-thickness (IMT) and reduced estimated glomerular filtration rate (e-GFR) [3]; furthermore, NGT ≥ 155 individuals exhibit higher left ventricular mass (LVM) values and higher prevalence of left ventricular hypertrophy (LVH), similar to impaired glucose tolerance (IGT) subjects and T2DM patients [4]. 

Notably, also left ventricular (LV) diastolic function is significantly worse in NGT ≥ 155 in comparison with NGT < 155 and similar to IGT and T2DM subjects. These observations are clinically relevant and underline the correlation between early alteration of glucose tolerance and myocardial damage. 

To improve clinical prognosis, it is fundamental to recognize early morphological and functional cardiac alterations even before the appearance of clinically evident reductions in LV performance. In accordance with this, in the last two decades myocardial deformation evaluation represents one of the most important innovations in the echocardiographic field because it considers many parameters that provide information about myocardial function beyond standard echocardiographic indices [5]. 

In particular, speckle tracking echocardiography (STE) is a second-level echocardiographic technique that allows semi-automatic quantification of myocardial global and regional deformation, for systolic and diastolic function, with an early detection of cardiac efficiency impairment [6]. STE allows us to track and display myocardial deformation global longitudinal strain (GLS) through its different layers. GLS has a prognostic value for CV events in diabetic patients with no history of CV complications and provides incremental prognostic value to clinical parameters, glycated haemoglobin (HbA1c), and diastolic function values in these patients [7,8,9,10]. Of interest, myocardial contractility is characterized by a higher deformation amplitude in the endocardial layer compared with the epicardial one, resulting in an endocardial/epicardial strain ratio (Endo/Epi strain) [11]. A detailed analysis of layer-specific mechanical changes in patients with glycol-metabolic disorders could provide a better knowledge of pathophysiological mechanisms underlying LV remodelling and subclinical myocardial damage in prediabetes and diabetic cardiomyopathy (DCM) [10]. 

In addition, myocardial work (MW) is a new dynamic non-invasive method that takes into account myocardial deformation and blood pressure (BP) values, thus overcoming the load dependency of LV ejection fraction (LVEF) and LV strain [11]. Global MW is a sensitive predictor of the subclinical LV systolic dysfunction in diabetic patients with preserved LVEF [12]. Moreover, a study conducted by Chen et al., demonstrated that hypertensive and diabetic patients had higher global myocardial wasted work (GWW) and lower global myocardial work efficiency (GWE) compared to controls, concluding that GMW may be a sensitive tool for the detection of subclinical changes in cardiac function in cardiometabolic diseases [13]. 

It is well known that in diabetic patients several pathophysiological factors, in particular metabolic abnormalities, inappropriate activation of the renin–angiotensin–aldosterone system (RAAS), and inflammation, may increase oxidative stress, which represents a common pathophysiological mechanism underlying CV diseases (CVD) and DCM [14]. Oxidative stress results from overproduction and decreased elimination of reactive oxygen species (ROS); the increased ROS production stimulates lipid peroxidation and protein carboxylation bringing the dysfunction of metabolic pathways and cellular components [15]. Nox-2, the catalytic core of NADPH oxidase enzyme, is the most important cellular source of ROS. The reaction of ROS with arachidonic acid on cell membranes leads to the production of isoprostanes, which in turn induces platelet activation, vasoconstriction, and activation of an inflammatory cascade [16,17]. These mechanisms may induce all the structural and functional cardiac changes characterizing DCM, such as myocardial interstitial fibrosis, cardiac stiffening with diastolic dysfunction, and finally, systolic dysfunction with clinically evident heart failure [18,19,20]. 

In accordance with this, chronic hyperglycaemia and insulin resistance (IR), which also characterize patients with NGT ≥ 155, by promoting mild chronic inflammation and oxidative stress, could play a predominant role in the association between higher one-hour postload plasma glucose levels and CV damage. In addition, it is conceivable that NGT ≥ 155 patients, characterized by a worse metabolic and inflammatory profile, show a greater oxidative stress than NGT < 155 similar to IGT and diabetic subjects. 

Therefore, it is a crucial issue to identify a possible association between increased oxidative stress and early myocardial alterations in order to understand the processes that lead to an increased CV risk before T2DM appearance. 

We designed this study to evaluate a possible correlation between oxidative stress validated markers (8-isoprostane and Nox-2) and myocardial deformation and efficiency parameters, obtained by STE, in NGT ≥ 155 patients, comparing them to NGT < 155 subjects, IGT, and newly diagnosed T2DM patients.

## 2. Materials and Methods

### 2.1. Study Population

We enrolled 100 newly diagnosed hypertensive Caucasian patients (61 men and 39 women, mean age 61.4 ± 10.7) participating in the Catanzaro Metabolic Risk Factors (CATAMERI) Study [21]. All subjects underwent physical examination and review of their medical history. Causes of secondary hypertension were excluded by appropriate clinical and biochemical tests. Other exclusion criteria were clinical evidence of heart failure or other CV complications, higher plasma levels of N-terminal pro-B-type natriuretic peptide (NT-proBNP), diagnosis of anaemia, history or malignant or chronic respiratory disease, endocrinological pathologies, malabsorption diseases, alcohol, drug, or smoking abuse. All subjects underwent anthropometrical evaluation with determination of weight, height, and body mass index (BMI). 

The ethics committee approved the protocol and informed written consent was obtained from all participants (code protocol number 2012.63). All of the investigations were performed in accordance with the principles of the Helsinki Declaration. 

### 2.2. Blood Pressure Measurement 

Clinical BP measurements were obtained according to current guidelines. Readings of BP were obtained in the left arm of the supine patients, after 5 min of rest, using a standard sphygmomanometer. BP values were the average of three measurements after a 10 min period of rest in the supine position. This evaluation was repeated on three separate occasions at least 2 weeks apart. Patients with a clinic systolic BP (SBP) > 140 mmHg and/or diastolic BP (DBP) > 90 mmHg were defined as hypertensive [22]. Pulse pressure (PP) values were obtained as the difference between systolic and diastolic BP measurements.

### 2.3. Laboratory Determination 

All laboratory measurements were performed after a fast of at least 12 h. A 75 g OGTT was performed with 0, 30, 60, 90, and 120 min sampling for plasma glucose and insulin. Glucose tolerance status was defined on the basis of OGTT using the World Health Organization (WHO) criteria. Plasma glucose was measured by the glucose oxidation method (Beckman Glucose Analyzer II; Beckman Instruments, Milan, Italy), and plasma insulin concentration was determined by a chemiluminescence-based assay (Roche Diagnostics). 

T2DM was defined according to the American Diabetes Association (ADA) criteria [23]. 

Insulin sensitivity was evaluated using the Matsuda index (insulin sensitivity index [ISI]), calculated as follows: 10.000/square root of [fasting glucose (millimoles per liter) × fasting insulin (milliunits per liter)] × [mean glucose3meaninsulin during OGTT]. The Matsuda index is strongly related to euglycemic-hyperinsulinemic clamp, which represents the gold standard test for measuring insulin sensitivity [24]. 

Triglycerides, total, LDL, and HDL cholesterol concentrations were measured by enzymatic methods (Roche Diagnostics, Mannheim, Germany). Values of estimated glomerular filtration rate (e-GFR) were calculated by using the equation proposed by investigators in the chronic kidney disease epidemiology (CKD-EPI) collaboration [25]. 

### 2.4. Serum Levels of Oxidative Stress Biomarkers 

Blood samples, obtained from fasted patients, were taken in tubes with separator gel and centrifuged at 4000 rpm for 15 min to obtain serum samples that were immediately stored at −80 °C.

Quantitative determination of the 8-isoprostane (ELISA kit Cayman Chemical, Ann Arbor, MI, USA) and Nox-2 (ELISA kit MyBioSource, San Diego, CA, USA) was performed with commercial ELISA immunoassays according to the manufacturer’s instructions. Values of 8-isoprostane were expressed as pg/mL; the lower detection limit of the assay was 0.8 pg/mL. Values of Nox-2 were expressed as nmol/L, the lower detection limit of the assay was 0.25 nmol/L. The coefficient of variation (CV %) was <9%.

### 2.5. Echocardiographic Measurements

Comprehensive 2D and Doppler echocardiography were performed using a commercially available ultrasound machine (Vivid E95, GE Healthcare, Horten, Norway) and performed according to the American Society of Echocardiography (ASE) recommendations [26]. Echocardiographic readings were made in random order by the investigator, who had no knowledge of patient’s clinical data. 

LVM was calculated using the formula proposed by Devereux et al. and corrected for body surface area (BSA), to derive the LVM index (LVMI) [27]. LV end-diastolic and end-systolic volumes and LVEF were calculated using biplane disk-summation algorithm [26], and the indexed by BSA. The values of all parameters were obtained as the average value of three consecutive cardiac cycles. 

### 2.6. 2D Speckle Tracking Echocardiography

A 2D speckle tracking analysis was retrospectively performed using vendor-specific 2D speckle tracking software (EchoPAC PC, version 113.0.5, GE Healthcare, Horten, Norway). Manual tracings of the endocardial border during end-systole in three apical views was performed to evaluate GLS.

The transmural variation of the longitudinal strain across the myocardial wall was calculated under the assumption of linear distribution. The endocardial and epicardial strains were calculated exactly on the endocardial and epicardial region of interest (ROI) borderlines, respectively. Subsequently, the values of transmural (not midwall), endocardial, and epicardial strain were obtained. The ratio of endocardial GLS to epicardial GLS was calculated using the endocardial GLS/Epicardial GLS (Endo/Epi) ratio for the assessment of the strain gradient (8). 

Additional indices of myocardial work were also calculated, in particular global constructive work (GCW) (myocardial work performed during LV shortening in systole and LV lengthening during the isovolumic relaxation phase); GWW (myocardial work performed during LV lengthening in systole and LV shortening in isovolumic relaxation phase); and GWE (constructive work divided by the sum of constructive and wasted work) [28]. 

Constructive work describes the net effect resulting from positive work (shortening) performed during systole and negative work (lengthening) performed during isovolumic relaxation. Wasted work describes the net effect resulting from negative work (lengthening) performed during systole plus positive work (shortening) during isovolumic relaxation. For the current analysis, LV segments with poor tracking or suboptimal image quality were excluded, as were subjects whose echocardiograms provided information on GLS and myocardial work in less than 17 segments. The software further provides a global work index (GWI = total work performed = area of the pressure-strain loop) and the global work efficiency (GWE = GCW/(GCW + GWW)) [28]. 

### 2.7. Statistical Methods 

Normally distributed data are reported as mean ± SD, and those not normally distributed as median and interquartile range. Variables having a positively skewed distribution were log transformed (lg10) before the correlational analysis. To test the differences among groups, analysis of variance (ANOVA) for clinical and biological data was performed, followed by the Bonferroni post hoc test for multiple comparisons. Chi-squared test was considered for categorical variables. Correlational analysis was carried out for the whole study population and according to different glucose tolerance groups. A linear correlation analysis was performed to compare GLS, GLS endo/epi ratio and GWE with different covariates including one-hour postload glucose, and oxidative stress biomarkers. Afterwards, variables achieving statistical significance were inserted in a multiple stepwise multivariate linear regression model to determine the independent predictor of GLS, GLS endo/epi ratio, and GWE. Data were considered significant at *p* <  0.05. All comparisons were performed using the statistical package SPSS 20.0 for Windows (SPSS Inc., Chicago, IL, USA).

## 3. Results

### 3.1. Study Population

Among 100 patients evaluated by OGTT, 54 showed NGT, 28 IGT, and 18 have T2DM. Considering the cut-off point of 155 mg/dL for one-hour postload plasma glucose, NGT subjects were stratified into two groups: 30 with NGT < 155 (55.6%), and 24 with NGT ≥ 155 (44.4%). 

In Table 1, the demographic, clinical, and biochemical characteristics of the study population are reported according to different metabolic states. 

There were no significant differences among groups regarding gender distribution and age. Anthropometric parameters, DBP, PP, heart rate (HR), total HDL and LDL cholesterol, and haemoglobin (Hb), were also not significantly different. From the first to the fourth group, there was a significant increase in SBP, triglyceride, and hs-CRP values as well as a reduction in e-GFR. 

Of interest, one-hour postload plasma glucose levels (*p* < 0.0001) and two-hour postload plasma glucose levels (*p* < 0.0001), as well as fasting (*p* < 0.0001), one-hour (*p* = 0.001) and two-hour (*p* < 0.0001) insulin values during OGTT significantly increased from the first to the fourth group. As expected, there was a worsening of insulin sensitivity accounting for the reduction in MATSUDA/ISI (*p* < 0.0001). 

Post hoc analysis by Bonferroni test confirmed that NGT ≥ 155 subjects have significantly worse insulin sensitivity as represented by a lower Matsuda index (*p* < 0.0001) and higher values of hs-CRP (*p* = 0.006), in comparison with NGT < 155, showing a metabolic and inflammatory profile similar to IGT individuals. 

### 3.2. Oxidative Stress Serum Parameters

Figure 1 and Figure 2 show serum levels of oxidative stress biomarkers (8-isorpostane, Nox-2), according to glucose tolerance status. 

There was a significant increase, from the first to the fourth group, in 8-isoprostane (*p* < 0.0001) and Nox-2 (*p* < 0.0001) serum levels, indicating a worsening in oxidative stress with the deterioration of metabolic status. In particular, 8-isoprostane (*p* < 0.0001) and Nox-2 (*p* < 0.0001) levels were significantly higher in the NGT ≥ 155 subjects compared to NGT < 155 group, without significant differences with IGT and diabetic patients.

### 3.3. Echocardiographic Parameters According to Glucose Tolerance

In Table 2, morphological and functional echocardiographic parameters of the study population according to different glycometabolic status are reported. 

From the first to the fourth group there was a significant increase in LVMI; diabetic subjects exhibited a worsening in both LVM (*p* < 0.0001) and LVMI (*p* < 0.0001) in comparison with NGT < 155, NGT ≥ 155 and IGT. In particular, NGT ≥ 155 patients showed significantly higher values of diastolic interventricular septum (dIVS) (*p* = 0.009), diastolic posterior wall (dPW) (*p* = 0.01) and LVMI (*p* = 0.026) in comparison with NGT < 155, but similar to IGT. By contrast, NGT ≥ 155 group compared with T2DM group, exhibited significantly lower LVMI (*p* = 0.009), while no significant differences were detected between the two groups as regards dIVS and dPW. Regarding the left ventricular end-diastolic diameter (LVEDD), there were no statistically significant differences between the four groups.

Regarding left global systolic function values evaluated as EF and stroke volume, significant differences were not detected among the four groups.

From the first to the fourth group, there was a progressive worsening of left global systolic function evaluated as myocardial deformation and GLS (*p* < 0.0001) (Figure 3). Furthermore NGT ≥ 155 subjects in comparison with NGT < 155 patients, showed a more compromised GLS (*p* < 0.0001), but similar to IGT. In addition, there was a progressive impairment in GLS endo values (*p* < 0.0001) and GLS endo/epi (*p* < 0.0001) proceeding from the NGT < 155 group to T2DM group. No differences were highlighted in GLS epi values among the four groups. 

Moreover, from the first to the fourth group there was a progressive reduction in GWE (Figure 4) (*p* < 0.0001) and a progressive increment in GWW (*p* < 0.0001). In detail, NGT ≥ 155 exhibited a worsening of GWE in comparison with NGT < 155 (*p* = 0.027), but similar to IGT and diabetic patients. 

In addition, a significant reduction in right systolic function was observed as demonstrated by lower values of TAPSE (*p* = 0.006) in NGT ≥ 155 group compared to NGT < 155; this parameter was also decreased in diabetic patients compared to the remaining groups (*p* < 0.0001). Values of left pulmonary systolic pressure (s-PAP) were similar in NGT ≥ 155, IGT and T2DM groups, with significant lower values in NGT < 155 (*p* < 0.0001). Moreover, TAPSE/s-PAP ratio significantly decreased in T2DM group compared to NGT < 155 (*p* < 0.0001) and NGT≥155 (*p* = 0.025), remaining similar to IGT. 

Finally, NGT≥155 had a significantly lower E/A ratio in comparison with NGT < 155 (*p* < 0.0001), but similar to IGT and T2DM patients. 

### 3.4. Correlation Analysis 

Correlation analysis was performed to test the correlation between echocardiographic parameters (GLS endo/epi ratio, GLS, and GWE) and different covariates (Table 3). 

GLS endo/epi ratio was significantly correlated with e-GFR (r = 0.280, *p* = 0.002), Matsuda/ISI (r = 0.541, *p* < 0.0001), and inversely correlated with one-hour glucose (r= −0.559, *p* < 0.0001), 8-isoprostane (r = −0.537, *p* < 0.0001), hs-CRP (r = −0.276, *p* = 0.003), Nox-2 (r = −0.551, *p* < 0.0001), E/e’ (r = −0.494, *p* < 0.0001), and LVMI (r = −0.401, *p* < 0.0001). When GLS was considered as a dependent variable, in the whole study population it was significantly correlated with e-GFR (r = −0.184, *p* = 0.034), Matsuda/ISI (r= -0.538, p<0.0001), Hb (r = −0195, *p* = 0.026), one-hour glucose (r = −0.564, *p* < 0.0001), 8-isoprostane (r = 0.445, *p* < 0.0001), hs-CRP (r = 0.370, *p* = 0.005), Nox-2 (r = 0.468, *p* < 0.0001), E/e’ (r = 0.473, *p* < 0.0001), and LVMI (r = 0.408, *p* < 0.0001).

Subsequently, GWE was considered as a dependent variable and was significantly correlated with e-GFR (r = 0.199, *p* = 0.023), Matsuda/ISI (r = 0.496, *p* < 0.0001), Hb (r = 0.347, *p* < 0.0001), one-hour glucose (r = −0.420, *p* < 0.0001), PP (r = −0.237, *p* = 0.009), 8-isoprostane (r = −0.279, *p* = 0.002), hs-CRP (r= −0.465, *p* < 0.0001), Nox-2 (r = −0.336, *p* < 0.0001), E/e’ (r= −0.359, *p* < 0.0001), and LVMI (r = −0.302, *p* = 0.001). 

Finally, variables achieving statistical significance were inserted in a stepwise multivariate linear regression model to determine the independent predictors of GLS endo/epi ratio, GLS, and GWE, respectively. In the whole study population, one-hour glucose was the major predictor of GLS endo/epi ratio explaining 31.3% of its variation (*p* < 0.0001) and Nox-2 added another 4.1%. Moreover, one-hour glucose was the major predictor of GLS explaining a 31.8% of its variation (*p* < 0.0001) and Matsuda/ISI added another 4.7% (*p* < 0.0001). Considering GWE as dependent variable, Matsuda/ISI was the main predictor justifying 24.6% (*p* < 0.0001) of its variation; Hb added another 7.8% and hs-CRP another 3.4%. When Matsuda/ISI was excluded from the model, hs-CRP resulted the major predictor of GWE variation justifying a 21.6% of its variation (*p* < 0.0001), one-hour glucose added a 7.3% (*p* < 0.0001) and Hb another 3.8% (*p* < 0.0001). The respective models justified a 35.4% for GLS endo/epi ratio, a 36.5% for GLS and 35.8% for GWE variation, respectively. 

## 4. Discussion

The results of this study are in agreement with previous results obtained by our research group suggesting that one-hour postload glucose ≥ 155 mg/dL is associated with a worse cardio-metabolic risk burden. Several studies show that one-hour postload glucose values allow us to identify subjects at a higher risk of developing T2DM and CVD, more efficaciously than two-hour postload glucose levels [2,3,4,29,30,31]. In the Israel Study of Glucose Intolerance, Obesity and Hypertension over a 33-year period, 945 individuals without diabetes completed an OGTT and were followed for all-cause mortality. Of interest, a one-hour glucose value >155 mg/dL predicts mortality even when the two-hour level was <140 mg/dL [30]. Moreover, in the Malmo project for prevention, conducted in 33346 apparently healthy, middle-aged men without diagnosis of diabetes, one-hour postload plasma glucose levels provided better prognostic yield than fasting blood glucose and two-hour postload plasma glucose, in predicting long-term CV morbidity and mortality [32]. Recently, in the Oulu Project Elucidating Risk of Atherosclerosis (OPERA) study, a population-based study consisting of 977 middle-aged subjects who underwent an OGTT, during a median follow-up of 19.8 years, one-hour glucose was a better long-term predictor of CV morbidity and mortality than fasting or two-hour glucose, independently of conventional CV risk factors [31].

The present study confirms that NGT ≥ 155 subjects, in comparison with NGT < 155, present a more compromised inflammatory profile, in particular higher hs-CRP levels, similar to that observed in IGT and T2DM subjects. Moreover, there was a progressive increase, from NGT ≥ 155 to T2DM, in TG levels. Dyslipidaemia is frequently associated with T2DM; in fact, in diabetes, many elements may influence blood lipid levels, because of the relationship between carbohydrates and lipids [33]. Furthermore, microvascular and microvascular complications, including CVD, are related with hyperglycaemia and dyslipidaemia in diabetic patients. 

Of interest, our study demonstrated an increased oxidative stress with the worsening of the metabolic profile; in fact, one of the novelties of our study is that a statistically significant increase in 8-isoprostane and Nox-2 levels was found in patients NGT ≥ 155 compared to NGT < 155, but similar to IGT patients. In the literature, there is growing evidence that oxidative stress has a crucial role in the pathological processes observed in prediabetes and T2DM [14,34]. ROS generated by Nox-2, inactivate nitric oxide (NO), which can interact with arachidonic acid with the consequent production of isoprostanes, including 8-isoprostane, favouring cell damage. In accordance with this, it’s plausible that increased Nox-2 levels in the NGT ≥ 155 subjects, led to an increase in 8-isoprostane levels; in fact, these patients have higher 8-isoprostane levels than NGT < 155 subjects and similar to IGT and diabetic subjects. Moreover, high levels of ROS are associated with aging rate; in fact, high levels of ROS are potentially critically for induction and maintenance of cell senescence process [35]. In our study, patients are in the age groups 58–65, and this could be another factor involved in the increase in ROS levels.

It has already been shown that oxidative stress plays an important role in the pathogenesis of CVD in diabetic subjects, by promoting endothelial dysfunction, characterized by reduced bioavailability of NO, with alterations in the vascular smooth muscle cells and cardiac damage such as LVH [36], but no study has ever evaluated the possible correlation between one-hour postload glucose ≥155 mg/dL and early myocardial alterations. 

The most clinically relevant information obtained from the present study is that NGT ≥ 155 subjects showed not only an increase in oxidative stress, but also a worse subclinical morpho-functional cardiac damage, similar to IGT and diabetic subjects, despite the patients being asymptomatic and with preserved LVEF. 

Recently multilayer strain analysis of LV mechanics in asymptomatic, hypertensive and diabetic patients revealed lower longitudinal strain in endo/layer and lower circumferential strain in endo and mid layer compared to healthy controls, which is also significantly correlated with HbA1c levels [10].

A study conducted by Bogdanovic et al., showed that, in asymptomatic diabetic patients with preserved LVEF, chronic hyperglycaemia exerted negative effects on the LV myocardium, by reducing GLS and GLS endo/epi ratio and highlighted early abnormalities even in the absence of overt symptoms [37]. In accordance with this, in our study we showed a progressive reduction in GLS and GLS endo/epi ratio in the enrolled population, proceeding from the NGT < 155 group to T2DM group, even if all patients were asymptomatic and showed a preserved LVEF. 

Moreover, in the stepwise multivariate linear regression model, the major predictors of GLS endo/epi ratio and GLS variability were one-hour glucose and Nox-2. This result can be explained by pathophysiological mechanisms of early histological and functional alterations in DCM. In accordance with this, glucotoxicity, insulin-resistance, oxidative stress and mitochondrial damage represent early steps in CV damage, promoting extracellular matrix turnover and alteration of the contractile properties of myofibrils in the cardiomyocytes, impairment in adenosine triphosphate (ATP) production and myocardial work efficiency [38]. Oxidative stress, DNA damage and degradation, and the reduction in mitochondria number and mitochondrial protein synthesis make cardiac mitochondria less able to produce ATP, thus myocardial fibres perform less, and ATP production is used for GWW instead to GWE; a phenomenon also caused by an increase in myocardial fibrosis and reduction in ventricular compliance. An excessive production of ROS has negative effects on cardiac contractility, myocardial work efficiency, and global cardiac performance; this contractile dysfunction is due to the influence that ROS have on the interruption of the calcium cycle and the harmful effects that they cause on cellular metabolism, ATP production, and constructive cardiac work [36]. 

As demonstrated in a study conducted by Aasum et al., changes in cardiac metabolism appear early in the diabetic progression [39].

Previous animal and human studies have demonstrated that obesity and T2DM are linked with increased myocardial O2 consumption (MVO2) and decreased cardiac efficiency [40]. T2DM and insulin resistance are associated with increased cardiac fatty acid oxidation and simultaneous decreased of glucose utilization [41]. Studies conducted in animal model demonstrated that obesity and insulin resistance rise the plasma fatty acid available and myocardial fatty acid uptake (MFAUp), which is responsible of a shift in substrate metabolism toward a preference for fatty acid utilization (MFAU) [42]. The increase in myocardial fatty acid oxidation (MFAO) and MVO_2_ lead to a reduced cardiac efficiency. Moreover, human studies confirmed that obesity is associated with alteration in myocardial substrate metabolism, structure and efficiency and that the addiction on myocardial fatty acid metabolism boost with the worsening of insulin resistance. 

It is known that myocardial oxygen demand is the amount of oxygen that the heart requires to maintain optimal function and myocardial oxygen supply is the amount of the oxygen provided to heart by the blood [43]. In our study, we demonstrated that haemoglobin was significantly and directly correlated with GWE justifying 3.8% of its variation. Moreover, the most interesting data that emerged from our analysis was that there was a progressive reduction in GWE, according to the glycometabolic status, in the presence of normal haemoglobin range values. 

Our data stress the concept that NGT≥155 patients present a reduced myocardial efficiency and early functional alteration of myocardial contractile fibres, before LVEF reduction and clinically evident heart failure.

The lack of progressive deterioration of endocardial strain may be explained by the relatively small sample size. We would like to emphasize that there was a consistent trend of LV layer-specific strain reduction from the endocardium to the epicardium, which is in agreement with previous studies that used layer-specific strain, demonstrating a myocardial gradient of strain, with a decreasing strain value from the endocardial to the epicardial layer in hypertensive and diabetic patients [44]. 

## 5. Conclusions

In conclusion, the present study confirms the predictive role of one-hour postload plasma glucose values on metabolic and CV risk, highlighting the importance of performing OGTT, taking into account not only two-hour but also one-hour postload glucose levels to improve the stratification of CV risk.

The novelty of our study is the demonstration of early functional alterations of myocardial contractile fibres in NGT ≥ 155 subjects, even before the reduction in LVEF; moreover, the impairment in myocardial performance was strongly correlated with increased oxidative stress. Therefore, it is important to reconsider the concept that NGT patients are at a low CV risk. The data obtained from our study not only confirm that NGT ≥ 155 subjects have metabolic and inflammatory profiles similar to IGT and diabetic patients, but also that they present cellular and subclinical organ damage as evidenced by increased levels of oxidative stress, and a reduction in global and multilayer strain and myocardial work efficiency. 

These data could play a central role in ongoing research on the association between early glycometabolic alterations and CV outcome.

## Figures and Tables

**Figure 1 nutrients-14-01299-f001:**
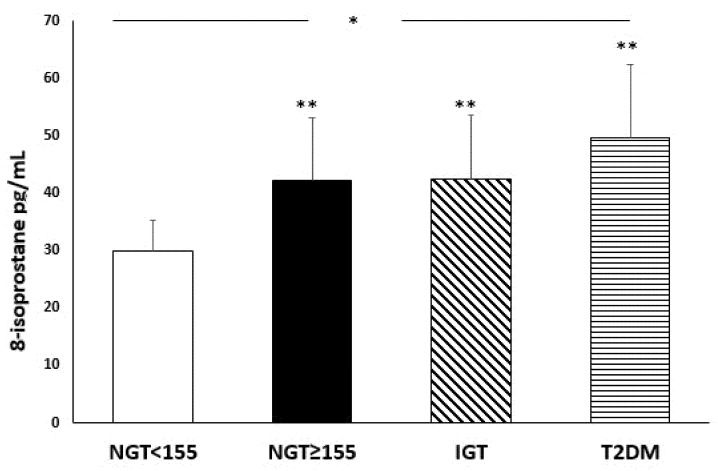
Serum levels of 8-isoprostane (pg/mL) in the four groups according to glucose tolerance status. * *p* < 0.0001 (ANOVA test); ** *p* < 0.0001 vs. NGT < 155 (Bonferroni post hoc test).

**Figure 2 nutrients-14-01299-f002:**
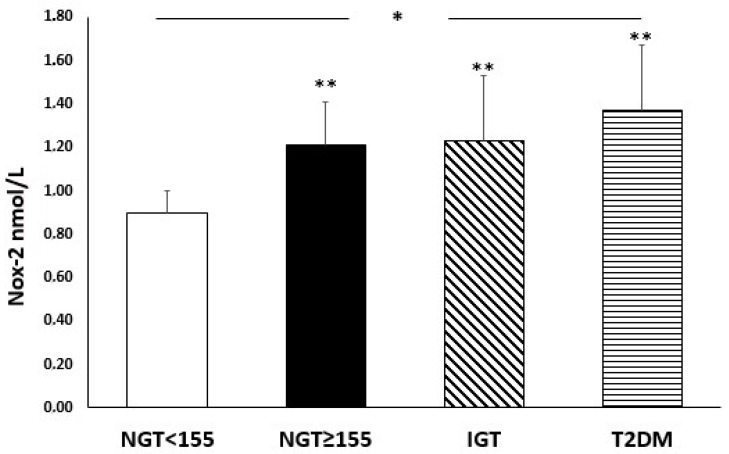
Serum levels of Nox-2 (nmol/L) in the four groups according to glucose tolerance status. * *p* < 0.0001 (ANOVA test); ** *p* < 0.0001 vs. NGT < 155 (Bonferroni post hoc test).

**Figure 3 nutrients-14-01299-f003:**
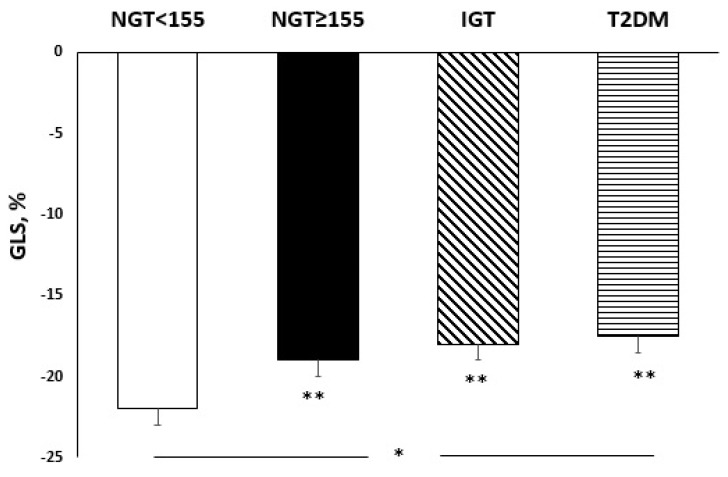
Global longitudinal strain (%) according to different groups of glucose tolerance. * *p* < 0.0001 (ANOVA test); ** *p* < 0.0001 vs. NGT < 155 (Bonferroni post hoc test).

**Figure 4 nutrients-14-01299-f004:**
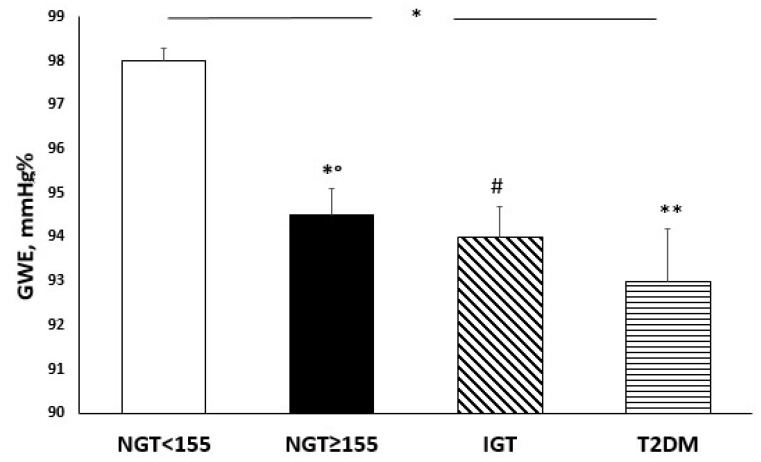
Global myocardial work values (mmHg%) according to different groups of glucose tolerance. * *p* < 0.0001 (ANOVA test); ** *p* < 0.0001 vs. NGT < 155, *° *p* = 0.016 vs. NGT < 155; # *p* = 0.001 vs. NGT < 155 (Bonferroni post hoc test).

**Table 1 nutrients-14-01299-t001:** Anthropometric, hemodynamic, and biochemical characteristics of the study population according to glucose tolerance status.

Variables	All (*n* = 100)	NGT < 155 (*n* = 30)	NGT ≥ 155 (*n* = 24)	IGT (*n* = 28)	T2DM (ND) (*n* = 18)	*p* ^a^
**Gender**, m/f	61/39	13/17	15/9	21/7	12/6	0.080
**Age**, years	61.4 ± 10.7	58.0 ± 12.2	65.0 ± 8.0	61.6 ± 11.2	62.2 ± 9.4	0.120
**BMI**, Kg/m^2^	30.5 ± 5.0	29.6 ± 5.7	31.6 ± 4.6	31.1 ± 5.1	29.7 ± 4.0	0.402
**SBP**, mmHg	134.4 ± 11.4	130 ± 13.0	134.6 ± 10.0	135.3 ± 9.0	139.8 ± 12.0	0.031
**DBP**, mmHg	79.9 ± 9.6	79.0 ± 8.8	81.2 ± 10.9	79.1 ± 9.8	80.8 ± 9.1	0.766
**PP**, mmHg	54.7 ± 12.2	51.1 ± 13.1	53.4 ± 13.2	56.2 ± 11.1	60.4 ± 8.8	0.061
**HR**, bpm	69.9 ± 6.2	69.2 ± 5.3	68.6 ± 5.9	69.6 ± 5.6	73.5 ± 7.8	0.054
**FPG**, mg/dL	99.2 ± 17.0	88.1 ± 11.7	95.9 ± 18.7	101.5 ± 12.5	118.4 ± 9.8	<0.0001
**1 h glucose**, mg/dL	180.7 ± 47.7	127.1 ± 24.1	183.9 ± 31.7	197.9 ± 21.5	238.8 ± 31.0	<0.0001
**2 h glucose**, mg/dL	154.6 ± 48.3	115.9 ± 29.6	125.0 ± 27.6	174.7 ± 14.6	227.4 ± 23.5	<0.0001
**Fasting insulin**, µU/mL	15.5 ± 6.0	11.2 ± 3.4	14.4 ± 4.3	18.7 ± 5.9	19.2 ± 6.6	<0.0001
**1h-insulin**, µU/mL	100.6 ± 50.8	79.6 ± 38.4	119 ± 31.1	113.8 ± 66.3	90.4 ± 50.7	0.001
**2h-insulin**, µU/mL	98.1 ± 42.0	76.8 ± 24.0	80.9 ± 18.5	119.7 ± 48.1	122.8 ± 52.2	<0.0001
**Matsuda**/ISI	50.6 ± 20.9	73 ± 15.3	49.5 ± 11.2	38.0 ± 13.4	34.6 ± 16.5	<0.0001
**Total cholesterol**, mg/dL	179.2 ± 39.0	189.7 ± 40.0	178.8 ± 35.3	169.9 ± 34.5	176.5 ± 47.0	0.280
**Triglyceride**, mg/dL	138.1 ± 61.1	115.8 ± 31.0	123.0 ± 54.1	132.5 ± 71.4	203.9 ± 46.7	<0.0001
**HDL**, mg/dL	47.9 ± 12.1	50.5 ± 12.4	50.3 ± 11.9	45.5 ± 10.1	44.3 ± 13.5	0.170
**LDL**, mg/dL	118 ± 36.7	124.9 ± 41.5	118.4 ± 33.5	111.4 ± 29.0	116.2 ± 43.5	0.576
**hs-CRP**, mg/L	3.2 ± 2.4	1.5 ± 1.1	3.4 ± 1.9	4.2 ± 3.0	4.3 ± 1.7	<0.0001
**e-GFR**, mL/min/1.73 m^2^	103.5 ± 25.7	119.6 ± 28.0	102.3 ± 20.0	93.2 ± 21.3	94.4 ± 23.1	<0.0001
**Hb**, g/dL	14.0 ± 1.5	14.0 ± 1.3	14.0 ± 1.6	13.4 ± 1.6	13.4 ± 1.8	0.464

NGT = Normal Glucose Tolerance; IGT = Impaired Glucose Tolerance; T2DM = Type 2 Diabetes Mellitus; BMI = Body Mass Index; SBP = Systolic Blood Pressure; DBP = Diastolic Blood Pressure; PP = Pulse Pressure; HR = Heart Rate; FPG = Fasting Plasma Glucose; HDL = high density lipoprotein; LDL = low density lipoprotein; hs-CRP = high-sensitivity C-reactive protein; e-GFR = estimated glomerular filtration rate; Hb = haemoglobin. Data are mean ± SD. ^a^ = Overall difference among groups (ANOVA).

**Table 2 nutrients-14-01299-t002:** Morpho-functional echocardiographic parameters of the study population according to glucose tolerance.

Variables	All (*n* = 100)	NGT < 155 (*n* = 30)	NGT ≥ 155 (*n* = 24)	IGT (*n* = 28)	T2DM (ND) (*n* = 18)	*p* ^a^
**LVDD,** cm	5.1 ± 0.4	5.1 ± 0.4	5.1 ± 0.4	5.2 ± 0.3	5.2 ± 0.4	0.347
**dIVS,** cm	1.22 ± 0.1	1.1 ± 0.1	1.2 ± 0.1	1.2 ± 0.1	1.3 ± 0.1	<0.0001
**dPW,** cm	1.0 ± 0.1	0.9 ± 0.1	1.0 ± 0.1	1.0 ± 0.1	1.0 ± 0.1	<0.0001
**LVM,** g	198.5 ± 37.2	172.7 ± 22.1	193.7 ± 30.3	209.8 ± 32.4	230.2 ± 42.6	<0.0001
**LVMI,** g/m^2^	109 ± 19.7	94.2 ± 10.8	107.3 ± 17.6	116.8 ± 15.7	124 ± 22.7	<0.0001
**Stroke volume,** mL	78.9 ± 16.4	77.6 ± 12.3	74.5 ± 16.0	82 ± 16.4	81.5 ± 22.2	0.325
**LVEF**%	61.1 ± 4.2	62.7 ± 4.4	61 ± 4.7	60.3 ± 3.3	60 ± 3.5	0.052
**GLS**%	−19.4 ± 2.4	−21.1 ± 1.8	−19.1 ± 2.2	−18 ± 1.4	−18.1 ± 2.1	<0.0001
**GLSendo**%	−21.6 ± 2.9	−24.0 ± 1.5	−21.7 ± 2.2	−20.1 ± 2.3	−19.7 ± 3.7	<0.0001
**GLSepi** %	−16.1 ± 2	−16.1 ± 1.3	−16.5 ± 2	−15.5 ± 2	−16.3 ± 2.8	0.340
**GLS endo/epi ratio**	1.34 ± 0.2	1.5 ± 0.1	1.3 ± 0.2	1.3 ± 0.1	1.2 ± 0.2	<0.0001
**GWI,** mmHg %	2045.7 ± 370.5	2096.6 ± 491.4	1954.6 ± 285.5	2068.4 ± 311.9	2047.4 ± 325.0	<0.0001
**GCW,** mmHg %	2114.2 ± 407.1	2457.5 ± 419.8	2074.4 ± 220.2	1944.9 ± 350.2	1858.4 ± 279.2	<0.0001
**GWW,** mmHg%	93 ± 46.7	55.6 ± 25.6	100.3 ± 36.6	106.3 ± 44.7	125.3 ± 51.3	<0.0001
**GWE,** mmHg %	95.5 (93–97.75) *	98.0 ± 0.3	94.5 ± 0.6	94.0 ± 0.7	93.0 ± 1.2	<0.0001
**TAPSE,** mm	21.6 ± 3.0	24.0 ± 2.8	21.5 ± 2.5	20.9 ± 2.3	19.2 ± 1.9	<0.0001
**s-PAP,** mmHg	31.9 ± 6.8	26.4 ± 5.7	33.1 ± 6.5	33.9 ± 5.4	36.3 ± 5.4	<0.0001
**TAPSE/s-PAP,** mm/mmHg	0.7 ± 0.2	0.9 ± 0.2	0.7 ± 0.2	0.6 ± 0.1	0.5 ± 0.1	<0.0001
**E,** m/s	0.7 ± 0.2	0.9 ± 0.2	0.7 ± 0.2	0.6 ± 0.1	0.7 ± 0.2	<0.0001
**A,** m/s	0.8 ± 0.2	0.8 ± 0.2	0.8 ± 0.1	0.8 ± 0.2	1.0 ± 0.2	<0.0001
**E/A**	0.9 ± 0.3	1.2 ± 0.3	0.9 ± 0.3	0.8 ± 0.1	0.7 ± 0.1	<0.0001
**E/e’**	12.9 ± 3.4	9.3 ± 2.0	13.2 ± 1.5	14 ± 2.2	16.2 ± 3.2	<0.0001
**LAVI,** mL/mq	33.9 ± 5.0	30.4 ± 4.5	34.2 ± 3.5	34.5 ± 4.9	38.7 ± 3.1	<0.0001

LVDD = Left Ventricular diastolic diameter; dIVSd = diastolic interventricular septum, dPW = diastolic posterior wall; LVM = Left Ventricular Mass; LVMI = Left ventricular Mass Index; LVEF = Left Ventricular Ejection Fraction; GLS = Global longitudinal strain; GLS endo = GLS Endocardial; GLS epi = GLS epicardial; GLS endo/epi ratio = GLS Endocardial/Epicardial strain ratio; GWI = Global work index; GCW = Global Constructive Work; GWW = Global myocardial wasted work; GWE = Global myocardial work efficiency, TAPSE = Tricuspid annular plane systolic excursion; s-PAP = systolic pulmonary arterial pressure; LAVI = left atrial volume index; E = Wave E; A = Wave A. Data are mean ± SD. * Log-transformed (ln) variables. ^a^ = Overall difference among groups (ANOVA).

**Table 3 nutrients-14-01299-t003:** Linear regression analysis between GLS endo/epi ratio, GLS, GWE, and different covariates in the whole study population.

	GLS Endo/Epi Ratio	GLS	GWE
	r/p	r/p	r/p
**e-GFR,** ml/min/1.73 m2	0.280/0.002	−0.184/0.034	0.199/0.023
**Matsuda/ISI**	0.541/<0.0001	−0.538/<0.0001	0.496/<0.0001
**Hb,** g/dL	0.029/0.388	−0.195/0.026	0.347/0.001
**1 h glucose**, mg/dL	−0.559/<0.0001	0.564/<0.0001	−0.420/<0.0001
**PP,** mmHg	−0.086/0.198	0.160/0.056	−0.237/0.009
**8-isoprostane,** pg/mL	−0.537/<0.0001	0.445/<0.0001	−0.279/0.002
**hs-CRP,** mg/L	−0.276/0.003	0.370/<0.0001	−0.465/<0.0001
**Nox-2,** nmol/L	−0.551/<0.0001	0.468/<0.0001	−0.336/<0.0001
**HR,** bpm	−0.069/0.249	0.140/0.082	−0.086/0.198
**E/e’**	−0.494/<0.0001	0.473/<0.0001	−0.359/<0.0001
**LVMI,** g/m^2^	−0.401/<0.0001	0.408/<0.0001	−0.302/0.001

eGFR = estimated glomerular filtration rate; PP = Pulse Pressure; hs-CRP = high-sensitivity C-reactive protein; HR = Heart Rate; LVMI = Left Ventricular Mass Index; Hb = haemoglobin; GLS = Global Longitudinal Strain; GLS endo/epi ratio = GLS Endocardial/Epicardial strain ratio; GWE = Global myocardial work efficiency.

## Data Availability

The data are not publicly available due to privacy.

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
