# Peer review of "Oxidative Stress and Left Ventricular Performance in Patients with Different Glycometabolic Phenotypes"

_nutrients, 2022, doi:10.3390/nu14061299_

Round 1

Reviewer 1 Report

In this study the authors evaluated a possible correlation between oxidative stress markers and myocardial deformation and efficiency parameters in normal glucose tolerant (NGT) subjects, as well as in subjects with impaired glucose tolerance (IGT) and T2DM patients. The study population, consisted of 100 newly diagnosed hypertensive Caucasian patients participating in the Catanzaro Metabolic Risk Factors Study, where the NGT subjects were stratified into two groups on the basis of an oral glucose-tolerance test: 30 with one-hour post-load glucose <155 mg/dl mg/dl and 24 with one-hour post-load glucose ≥ 155 mg/dl. Serum values of oxidative stress markers (8-isoprostane and Nox-2) were determined by ELISA tests. Cardiac function was assessed by two-dimensional echocardiography, while speckle tracking echocardiography (SPE) was performed retrospectively to assess subclinical myocardial damage. Additional indices of myocardial work and efficiency were also calculated.

The results of the present study confirm previous results from this research group, suggesting that one-hour post-load glucose ≥ 155 mg/dl is associated with a clear cardio-metabolic risk burden. In this study, however, they also show that NGT≥155 subjects present early functional alterations of myocardial contractile fibres, even before the reduction of left ventricular function assessed by two-dimensional echocardiography (stroke volume, left ventricular ejection fraction). Furthermore, the impairment in myocardial performance was strongly correlated with increased oxidative stress. The authors conclude that it is important to reconsider the concept that NGT patients are at low CV risk.

This study is based on a limited study population, which excludes any examination of possible gender differences in the results obtained. It is well conducted and well written however, with the exception that it is not always easy to find out when reading the tables which groups are statistically different from each other. The results are interesting, though, and this reviewer finds few points to criticize.

Regression analysis showed that one-hour glucose and Nox-2 were the major predictors of myocardial damage (GLS endo/epi ratio and GLS variability, and they suggest that glucotoxicity, insulin-resistance, oxidative stress, mitochondrial damage and energy wasting are involved. However, no direct evidence is provided to support this notion.

A very intriguing finding, in this reviewer’s mind, was that analysis of the present data showed a progressive reduction of global myocardial work efficiency (GWE) in accordance with the decline in glycometabolic status. Thus, finding is in line with pre-clinical studies in obese [Boardman N et al. Am J Physiol Heart Circ Physiol. 2009;296(5):H1373-9] and diabetic mice [How OJ et al. 2006, Diabetes, 55(2), 466–473], which demonstrate how deterioration of metabolic homeostasis (increased fatty acid oxidation at the expense of glucose) during these conditions impacts negatively on cardiac efficiency. Reduction of cardiac efficiency in obesity and diabetes is often neglected and therefore deserves to be included in the discussion, preferably with more focus more on insulin sensitivity/insulin resistance and how the consequent alterations in myocardial substrate supply and utilization affects cardiac efficiency. In this context it should also be noted that changes in myocardial substrate metabolism in diabetic animals precede functional deterioration [Aasum E et al. 2003 Diabetes;52(2):434-41].

Minor:

What does pa = 0.080* in Table 1 mean?

Line 294-295: “In detail, 294 NGT≥155 exhibited a worsening of GWE in comparison with NGT≥155 (p=0.027).” Should probably read: In detail, 294 NGT≥155 exhibited a worsening of GWE in comparison with NG<155 (p=0.027).

Line 342-342: …” hs-CRP resulted the major predictor…”

Iine 352: ….”than one-hour postload glucose levels”….Should probably read: ….than twoe-hour postload glucose levels.

Author Response

Reviewer 1

C1: In this study the authors evaluated a possible correlation between oxidative stress markers and myocardial deformation and efficiency parameters in normal glucose tolerant (NGT) subjects, as well as in subjects with impaired glucose tolerance (IGT) and T2DM patients. The study population, consisted of 100 newly diagnosed hypertensive Caucasian patients participating in the Catanzaro Metabolic Risk Factors Study, where the NGT subjects were stratified into two groups on the basis of an oral glucose-tolerance test: 30 with one-hour post-load glucose <155 mg/dl mg/dl and 24 with one-hour post-load glucose ≥ 155 mg/dl. Serum values of oxidative stress markers (8-isoprostane and Nox-2) were determined by ELISA tests. Cardiac function was assessed by two-dimensional echocardiography, while speckle tracking echocardiography (SPE) was performed retrospectively to assess subclinical myocardial damage. Additional indices of myocardial work and efficiency were also calculated.

The results of the present study confirm previous results from this research group, suggesting that one-hour post-load glucose ≥ 155 mg/dl is associated with a clear cardio-metabolic risk burden. In this study, however, they also show that NGT≥155 subjects present early functional alterations of myocardial contractile fibres, even before the reduction of left ventricular function assessed by two-dimensional echocardiography (stroke volume, left ventricular ejection fraction). Furthermore, the impairment in myocardial performance was strongly correlated with increased oxidative stress. The authors conclude that it is important to reconsider the concept that NGT patients are at low CV risk.

This study is based on a limited study population, which excludes any examination of possible gender differences in the results obtained. It is well conducted and well written however, with the exception that it is not always easy to find out when reading the tables which groups are statistically different from each other. The results are interesting, though, and this reviewer finds few points to criticize.

Regression analysis showed that one-hour glucose and Nox-2 were the major predictors of myocardial damage (GLS endo/epi ratio and GLS variability, and they suggest that glucotoxicity, insulin-resistance, oxidative stress, mitochondrial damage and energy wasting are involved. However, no direct evidence is provided to support this notion.

A very intriguing finding, in this reviewer’s mind, was that analysis of the present data showed a progressive reduction of global myocardial work efficiency (GWE) in accordance with the decline in glycometabolic status. Thus, finding is in line with pre-clinical studies in obese [Boardman N et al. Am J Physiol Heart Circ Physiol. 2009;296(5):H1373-9] and diabetic mice [How OJ et al. 2006, Diabetes, 55(2), 466–473], which demonstrate how deterioration of metabolic homeostasis (increased fatty acid oxidation at the expense of glucose) during these conditions impacts negatively on cardiac efficiency. Reduction of cardiac efficiency in obesity and diabetes is often neglected and therefore deserves to be included in the discussion, preferably with more focus more on insulin sensitivity/insulin resistance and how the consequent alterations in myocardial substrate supply and utilization affects cardiac efficiency. In this context it should also be noted that changes in myocardial substrate metabolism in diabetic animals precede functional deterioration [Aasum E et al. 2003 Diabetes;52(2):434-41].

R1. We thanks the reviewer for the constructive comments. We are agree with the reviewer that  the study is based only on 100 subjects and this is a study limitation . Moreover, in the discussion we further discussed the reduction of cardiac efficiency in obesity and diabetes.

Minor

C1. What does pa = 0.080* in Table 1 mean?

R1. aOverall difference among groups (ANOVA).

C2. Line 294-295: “In detail, 294 NGT≥155 exhibited a worsening of GWE in comparison with NGT≥155 (p=0.027).” Should probably read: In detail, 294 NGT≥155 exhibited a worsening of GWE in comparison with NG<155 (p=0.027).

R2. We corrected the mistake.

C3. Line 352: ….”than one-hour postload glucose levels”….Should probably read: ….than two-hour postload glucose levels.

R3. Thanks, we corrected the mistake.

Reviewer 2 Report

Manuscript # 1629626

In the current article titled “ Oxidative stress and left ventricular performance in patients with different glycometabolic phenotypes” authors have evaluated the role of one-hour post-load plasma glucose determination as a prognostic measure for cardiovascular disease. Authors have determined the association of oxidative stress and altered glycometabolic status with altered cardiac dysfunction (altered ventricular function) as a key determinant in the development of cardiovascular disease including diabetes. This is a very interesting piece of work with a small samples size and has added new prospects in the prognosis of cardiovascular disease. I have the following comments: 

Comments

  • Authors have determined the early alteration of cardiac function including increased oxidative stress, reduced global and multilayer strain (from endocardial to epicardial layer) and myocardial work efficiency in one-hour post-load plasma glucose (NGT≥ 155 mg/dl) before the onset of clinically evident heart failure symptoms including reduction of LVEF. This is an interesting observation study in the prognosis of cardiovascular disease. Authors have also evaluated the reduced global cardiac efficacy even in normal haemoglobin, HDL and LDL range which is an interesting observation to determine the cardiac function before the onset of disease and prognosis of CVD. 
  • Although authors have evaluated the role of oxidative stress in the development of left ventricular dysfunction in correlation with glycometabolic phenotype. However, authors have used study subjects in the age groups 58-65 where ageing could be another factor involved in this. Because ageing involved the accumulation of ROS and reduced metabolic activities. Thus, authors should discuss these points. 
  •     Authors have stated the role of oxidative stress and Inflammation in cardiac dysfunction; however, they have measured only a few markers of oxidative stress (two markers), they should use other oxidative stress markers along with NOX-2 and 8-isoprostane because diabetes and glucose-related pathological conditions show changes in metabolic profiles along with inflammatory chemokines and cytokines. Thus, authors should discuss different cytokines and oxidative stress markers. Only two markers cannot be prognostic markers.
  • Although authors have included male and female genders in their study subject. However, they have not discussed the changes in different parameters in male and female separatory because the pathological response in high glucose-related disease conditions is different for both sexes. 
  • The authors have used a very small sample size. They should use larger samples size and include lower age groups subjects ( 30-50 yrs age) because the onset of glucose-related pathology starts at early to mid-age of individuals.
  • Authors have found changes in triglycerides in all four groups which is increasing from NGT≥ 155mg/dl to T2DM compared with NGT≤155mg/dl. However, no changes in HDL and LDL. Authors should discuss changes in these metabolic parameters in the prognosis of diabetes and CVD.

Author Response

Reviewer 2

Authors have determined the early alteration of cardiac function including increased oxidative stress, reduced global and multilayer strain (from endocardial to epicardial layer) and myocardial work efficiency in one-hour post-load plasma glucose (NGT≥ 155 mg/dl) before the onset of clinically evident heart failure symptoms including reduction of LVEF. This is an interesting observation study in the prognosis of cardiovascular disease. Authors have also evaluated the reduced global cardiac efficacy even in normal haemoglobin, HDL and LDL range which is an interesting observation to determine the cardiac function before the onset of disease and prognosis of CVD.

Although authors have evaluated the role of oxidative stress in the development of left ventricular dysfunction in correlation with glycometabolic phenotype. However, authors have used study subjects in the age groups 58-65 where ageing could be another factor involved in this. Because ageing involved the accumulation of ROS and reduced metabolic activities. Thus, authors should discuss these points.

We thanks the reviewer for the constructive comments. The reviewer is right in saying that ROS accumulation is also associated with aging, in fact we discussed this point in the discussion.

Authors have stated the role of oxidative stress and Inflammation in cardiac dysfunction; however, they have measured only a few markers of oxidative stress (two markers), they should use other oxidative stress markers along with NOX-2 and 8-isoprostane because diabetes and glucose-related pathological conditions show changes in metabolic profiles along with inflammatory chemokines and cytokines. Thus, authors should discuss different cytokines and oxidative stress markers. Only two markers cannot be prognostic markers.

We thanks the reviewer for the comments. Nox-2 and 8-isoprostane markers are the most validated biomarkers for the study of oxidative stress. This is a cross sectional study, we performed association between oxidative stress biomarkers and different covariates.  Moreover, the two markers are not prognostic markers.

Although authors have included male and female genders in their study subject. However, they have not discussed the changes in different parameters in male and female separatory because the pathological response in high glucose-related disease conditions is different for both sexes.

We thanks the reviewer for the comments. In our study, we included male and female genders, by the way female sample size is very small and there was not statistical significance between genders. Moreover, the gender did not enter into the correlation analysis.    

The authors have used a very small sample size. They should use larger samples size and include lower age groups subjects (30-50 yrs age) because the onset of glucose-related pathology starts at early to mid-age of individuals.

We thanks the reviewer, we are agree we used a small sample size, but we enrolled subjects in a wide range of ages, in fact 25% of patients are in age group 30-55 years age. 

Authors have found changes in triglycerides in all four groups which is increasing from NGT≥ 155mg/dl to T2DM compared with NGT≤155mg/dl. However, no changes in HDL and LDL. Authors should discuss changes in these metabolic parameters in the prognosis of diabetes and CVD.

Thanks to reviewer, we discussed the changes in triglycerides and the association with diabetes and CVD in discussion.